

# GPEP v1.0: a Geospatial Probabilistic Estimation Package to support Earth Science applications

Guoqiang Tang[1], Andrew W. Wood[1,2], Andrew J. Newman[3], Martyn P. Clark[4], Simon Michael Papalexiou[5]

[1]Climate and Global Dynamics, National Center for Atmospheric Research, Boulder, Colorado, United States

[2]Civil and Environmental Engineering, Colorado School of Mines, Golden, Colorado, United States

[3]Research Applications Laboratory, National Center for Atmospheric Research, Boulder, Colorado, United States

[4]Centre for Hydrology, University of Saskatchewan, Canmore, Alberta, Canada

[5]Department of Civil Engineering, University of Calgary, Alberta, Canada

*Correspondence to*: Guoqiang Tang (guoqiang@ucar.edu)

**Abstract.** Ensemble geophysical datasets are foundational for research to understand the Earth System in an uncertainty-aware context, and to drive applications that require quantification of uncertainties, such as probabilistic hydro-meteorological estimation or prediction. Yet ensemble estimation is more challenging than single-value spatial interpolation, and open-access routines and tools are limited in this area, hindering the generation and application of ensemble geophysical datasets. A notable exception in the last decade has been the Gridded Meteorological Ensemble Tool (GMET), which is implemented in FORTRAN and has typically been configured for ensemble estimation of precipitation, mean air temperature, and daily temperature range, based on station observations. GMET has been used to generate a variety of local, regional, national and global meteorological datasets, which in turn have driven multiple retrospective and real-time hydrological applications. Motivated by an interest in expanding GMET flexibility, application scope and range of methods, we have developed a Python-based Geospatial Probabilistic Estimation Package (GPEP) that offers GMET functionality along with additional methodological and usability improvements, including variable independence and flexibility, an efficient alternative cross-validation strategy, internal parallelization, and the availability of the scikit-learn machine learning library for both local and global regression. This paper describes GPEP and illustrates some of its capabilities using several demonstration experiments, including the estimation of precipitation, temperature, and snow water equivalent ensemble analyses on various scales.



## 1 Introduction

Meteorological datasets are essential for hydrometeorological and climate analysis and a wide range of related applications, from hydrometeorological forecasting to century-scale water security studies. Numerous gridded meteorological datasets exist based on a variety of estimation approaches, including the spatial interpolation of ground stations (Daly et al., 1994; Harris et al., 2020; Livneh et al., 2015; Maurer et al., 2002), remote sensing measurements from satellite sensors and weather radars (Huffman et al., 2007; Joyce et al., 2004; Shen et al., 2018; Zhang et al., 2016), and atmospheric and Earth System modeling (Gelaro et al., 2017; Hersbach et al., 2020; Kobayashi et al., 2015; Muñoz-Sabater et al., 2021). Among these datasets, those based on ground station observations offer the most accurate and longest meteorological observations and are thus often used in the production of high-quality regional, national, and global gridded datasets. Station observations may be the sole input to the datasets, along with geophysical features that aid in a 'smart interpolation' to account for terrain and other influences or they may be used for bias correction of remote sensing and model estimates, or as the calibration reference for multi-source merging (Baez-Villanueva et al., 2020; Beck et al., 2019; Sun et al., 2018).

Methods for the spatial interpolation of station observations range in complexity from simpler strategies such as Thiessen polygons, distance-based weighting, linear interpolation, and nearest neighbour selection, to more complex procedures such as Kriging interpolation, geographically-weighted regression (GWR), and machine learning techniques. Many widely used deterministic meteorological datasets are produced using these methods or their variants, such as the Global Precipitation Climatology Centre (GPCC) dataset (Schamm et al., 2014) and the Climatic Research Unit gridded Time Series (CRU TS) dataset (Harris et al., 2020). Yet spatial interpolation is an imperfect process that leads to ubiquitous uncertainties in gridded meteorological datasets. Few meteorological datasets provide explicit analytical uncertainty estimates, and even fewer provide probabilistic or ensemble estimates, members of which can be advantageous in quantifying uncertainties and characterizing extreme events (Tang et al., 2023). To address this problem, several recent studies have developed station-based ensemble meteorological datasets, including the HadCRUT4 global temperature dataset (Morice et al., 2012), the Spatially COherent Probabilistic Extended Climate dataset (SCOPE Climate) in France (Caillouet et al., 2019), the ensemble precipitation and temperature datasets in the United States and parts of Canada (Newman et al., 2015, 2019, 2020), the Ensemble Meteorological Dataset for North America (EMDNA; Tang et al., 2021), and the Ensemble Meteorological Dataset for Planet Earth (EM-Earth; Tang et al., 2022). Several deterministic datasets such as the Europe-wide E-OBS (Haylock et al., 2008; Cornes et al., 2018) and Canadian Precipitation Analysis (CaPA; Mahfouf et al., 2007; Fortin et al., 2015; Khedhaouiria et al., 2020) also offer probabilistic realizations. In addition to these station-based datasets, there are also reanalysis ensembles such as ERA5 Ensemble of Data Assimilations (Hersbach et al., 2020) and satellite-based ensemble generation methods such as the satellite rainfall error model (Hossain & Anagnostou, 2006; Hartke et al., 2022) which are beyond the scope of this study.





However, the rise of ensemble meteorological datasets also brings new challenges or amplifies existing ones. First, like many
other historical datasets, ensemble datasets are often built on open-access station collections, with fixed periods and resolutions
and limited variables, which may not be updated routinely once the production is finished. Second, ensemble datasets often
have large data sizes increasing with the number of members, posing challenges in downloading, storage, and processing.
Third, ensemble estimation methods generally have much higher complexity compared to single-value spatial interpolation
methods, making it difficult for researchers and practitioners to produce their datasets following dataset and method description
publications. Therefore, open-access tools for creating ensemble meteorological datasets are equally important and sometimes
more useful to the community compared to public datasets. Several such spatial interpolation tools are available in various
stages of development, such as the Topographically InformEd Regression (TIER; Newman & Clark, 2020), GStatSim (MacKie
et al., 2022), TFInterpy (Chen & Zhong, 2022), multiscale GWR (MGWR; Oshan et al., 2019), but well-tested tools for
meteorological ensemble estimation remain rare. A notable exception is the Gridded Meteorological Ensemble Tool (GMET:
https://github.com/NCAR/GMET) which can be used to generate ensemble meteorological analyses (i.e., gridded surface
forcings) using the locally-weighted spatial regression method outlined in Clark & Slater (2006). After an initial FORTRAN
development effort (Newman et al., 2015), GMET has been further refined and expanded in the course of sequential application
projects, producing a number of regional to continental datasets (Bunn et al., 2022; Liu et al., 2022; Longman et al., 2019;
Newman et al., 2015, 2019, 2020; Wood et al., 2021).
Successful GMET applications to date motivated interest in enhancements to allow for a broader range of uses and available
methods. GMET's Fortran basis enables it to be computationally efficient and fast, but is more cumbersome for adding or
linking to new methodological modules than the widely used scripting and programming language Python, for which many
relevant method libraries exist, particularly including machine learning (ML) techniques. In addition, GMET's development
to date has only afforded a subset of the potential user control over implementation choices, and some settings that would be
required for more flexible implementation are currently hardwired. For instance, the most common application is to generate
ensembles of precipitation, mean air temperature, and air temperature range, and certain assumptions, functions, and settings
specific to precipitation and temperature must be changed in the code if other variables are of interest. Future development to
enhance the FORTRAN GMET toward greater flexibility and user control is a viable option, but we view Python as providing
a more convenient and extensible development environment and one that can engage a potentially larger community of
contributors. The major downside of pursuing future development in Python relative to FORTRAN is its relatively slower
computational speed of Python, a tradeoff that we view as being acceptable given the benefits.
We have thus developed the Python-based Geospatial Probabilistic Estimation Package (GPEP). GPEP includes and expands
upon most of the current functionalities of FORTRAN GMET, bringing new methodological and usability enhancements.
These include (1) a flexible and configurable user control for input/output variables, run parameters, predictors, and weight





functions; (2) options for using basic ML techniques for local and global regression; (3) an alternative, efficient approach for
cross-validation; and (4) more flexible input formatting, especially for dynamic gridded predictor inputs. GPEP draws from
and formalizes some functions that were previously applied in the production of the continental EMDNA (Tang et al., 2021)
and the global EM-Earth (Tang et al., 2022) datasets, while mimicking GMET functionality (such as cross-validation and
usage of both static and time-variant predictor information) from Bunn et al. (2022).
GPEP is a powerful tool for both research and applications of deterministic and ensemble distributed geophysical analysis
estimation, including the production of meteorological datasets to support retrospective and real-time modeling on various
scales. This paper summarizes the GMET methodology and GPEP enhancements and illustrates some of its capabilities using
several experimental applications.

## 95    2 GMET methodology

The core GMET methodology for probabilistic meteorological ensemble analyses assumes that the estimate of a
meteorological variable at a specific time and location can be described by a parametric probability distribution. For mean air
temperature and daily temperature range, the normal distribution is used by GMET in the form of $X \sim N(\mu, \sigma^2)$ where $\mu$ and
$\sigma$ are the mean value and standard deviation, respectively. Ensemble estimates can be obtained by sampling from the normal
distribution. For variables such as precipitation with skewed distributions, transformation methods such as Box-Cox are
applied to convert variables into Normal space.
For GMET, $\mu$ is represented by the deterministic gridded estimates obtained from locally weighted linear regression (LWLR),
using static terrain-related predictors such as latitude, longitude, elevation, topographic slope, and aspect (as in Clark & Slater,
2006 and Newman et al, 2015). GMET version 2.0 added the ability to use dynamic predictors such as precipitation and
temperature from atmospheric models to further improve the accuracy of gridded estimates (Bunn et al., 2022), as well as a k-
fold (including leave-one-out) cross-validation option to enable the use of predictive rather than calibration uncertainty in
ensemble generation. Cross-validation is also a critical option for predictor screening and selection. $\sigma$ is the uncertainty of
gridded regression estimates based either on the standard error of the regression or the prediction error (e.g., root mean squared
error from cross-validation). For the intermittent variable precipitation, GMET uses a locally-weighted logistic regression to
estimate the probability of precipitation (POP) to enable its probabilistic estimation: i.e., the binary probability of the event (0
or 1) is regressed against the static and/or dynamic predictors, which are also used in a precipitation amount regression.
GMET then generates distributed, spatiotemporally correlated random fields (SCRFs) that are used to sample the distributed
regression models, generating ensembles that each maintain the spatial and temporal correlation structures of the input
variables (Newman et al., 2015). For SCRF, the spatial correlation length is used to represent the spatial correlation structure





over the entire domain, the lag-1 auto-correlation of temperature and the cross-correlation between precipitation and daily
temperature range are used to represent the temporal correlation structure and intervariable relationship (Equation 1).
$\{R_{t,T} = \rho_{lag-1}R_{t-1,T} + \sqrt{1 - \rho_{lag-1}^2}R_{t-1,T} \quad R_{t,P} = \rho_{cross}R_{t,TR} + \sqrt{1 - \rho_{cross}^2}R_{t-1,P}$ (1)

where t and t-1 are the current and previous time steps, respectively. $R_T$, $R_{TR}$ and $R_P$ are 2-dimensional SCRFs of mean air
temperature, and precipitation, respectively. $\rho_{lag-1}$ is the lag-1 auto-correlation of temperature. $\rho_{cross}$ is the cross-correlation
between precipitation and daily temperature range. For t=0, the SCRF is generated for each variable based only on the spatial
correlation structure. The spatial correlation length, $\rho_{lag-1}$ and $\rho_{cross}$ can be estimated from station observations.
After obtaining $\mu$, $\sigma$, the POP and SCRF, GMET can generate any number of ensemble members. Let $R$ be the random number
from the SCRF for a specific location and time step, the probabilistic estimate ($x_T$) for temperature variables is:
$x_T = \mu_T + R \cdot \sigma_T$ (2)
For precipitation, non-zero values are generated in proportion to the POP. Let $F_N(y)$ be the cumulative density function (CDF)
of the standard normal distribution and $F_N(R)$ is the cumulative probability corresponding to the random number $R$. If $p_0$ is
the probability of an event, the event occurs only when $F_N(R)$ is larger than $p_0$, for which we need to calculate the scaled
cumulative probability of precipitation ($p_{cs}$):
$p_{cs} = \frac{F_N(R) - p_0}{1 - p_0}$ (3)
The probabilistic estimate is expressed similarly to Equation (2):
$y = \{0 \quad if \quad F_N(R) \leq p_0 \qquad \mu_P + F_N^{-1}(p_{cs}) \cdot \sigma_P \quad if \quad F_N(R) > p_0$

133 (4)

where $y$ is the precipitation in the Normal space and $F_N^{-1}(p_{cs})$ is the random value corresponding to $p_{cs}$. $y$ needs to be back-
transformed to obtain the final precipitation values ($x_P$). Details of the GMET methodology are introduced in previous
development and dataset studies (e.g., Clark & Slater, 2006; Newman et al., 2015; Tang et al., 2021; Bunn et al., 2022).





## 3 GPEP

GPEP offers many methodological (Table 1) and usability (Table 2) features that expand on GMET, and these are described in Sections 3.1 and 3.2, respectively. Like many software tools, GMET was first written for a specific application, and GPEP now generalizes a number of the hard-coded options to provide for broader usage.

### 3.1 Methodological improvements

Here we introduce some major methodological improvements of GPEP compared to GMET. These changes enhance GPEP's flexibility as a tool not only for dataset production but also for scientific research aimed at achieving higher estimation accuracy or comparing the performance of different methodological strategies.

**Variable selection flexibility:** The original GMET code was implemented to estimate precipitation, mean daily air temperature (Tmean), and daily temperature range (Trange), although it has also been used to estimate only precipitation. The spatial regression method and design, however, are applicable to arbitrary spatio-temporal variables, thus GPEP brings the variable selection and associated details into the user control ('configuration') file. This versatility enables GPEP to generate ensemble analyses s for other variables; in the Earth Science or geophysical context these might include other meteorological variables such as radiation, wind speed, humidity, and air pressure, which are commonly required for hydrological models, or even hydrological variables for which observations or other analyses exist, such as snow water equivalent (SWE).

**Spatial interpolation:** GMET supported only locally weighted linear and logistic regression, whereas GPEP expands the options beyond these two basic capabilities to also support any supervised learning method from the scikit-learn package (Pedregosa et al., 2011) that can use the *fit* function to train the model and use the *predict/predict_proba* to predict the output. Such techniques include ridge regression and classification, BayesianRidge regression, Lasso regression, ElasticNet regression, among others, for locally weighted regression, and regressors and classifiers of random forest (RF), multi-layer perceptron, support vector machine, among others, for global regression. Global regression builds one model for the entire study domain at every time step, which is far more efficient than the local regression methods, whereas users need to caution that global regression may have degraded accuracy compared to local regression which needs in-depth investigation for case studies. Users can define the method for continuous and classification regression and define model parameters following scikit-learn formats in the configuration file.

**Uncertainty estimation:** GMET has the option to use a standard k-fold cross-validation to obtain the uncertainty of each grid cell specific regression estimates, where the number of folds is specified by the user. The use of k-fold cross-validation increases the computational demand in proportion to the number of folds, which was feasible in GMET but not in GPEP, due to its slower speed and relatively costlier operation. Consequently, GPEP adopts an alternative cross-validated uncertainty



estimation strategy: (1) obtaining regression estimates at all stations points, using leave-one-out validation for local regression
and N-fold cross-validation for global regression; and (2) interpolating the resulting root mean square error from the station
points to each grid cell using a distance weighted (i.e., locally weighted) averaging. The GPEP method achieves generally
similar uncertainties with the standard method at less computational cost. The similarity of the two error estimation outcomes,
however, will depend on the nature of the station and grid datasets being used.
**Spatial correlation length:** This parameter is critical for generating SCRFs for ensemble member generation. GMET requires
prescribed length values, whereas GPEP supports either user-specified correlation lengths, or else can infer the correlation
length from raw station inputs (a data-driven option). Users can also set various thresholds for the correlation calculation. For
example, a positive threshold such as 10 mm/d can be used to focus only on heavy precipitation. With the data driven option,
users need to ensure that the input data length is enough for robust estimation of the correlation; the prescribed option is useful
for smaller datasets (such as an operational forecast application) that are inadequate to define such correlation lengths.
**Static and dynamic predictors:** GMET uses a fixed grid for both the static and dynamic predictors, has a hard-coded default
list of static predictors, and uses the same predictors for all target variables (with a minor exception of dropping slope from
low-relief prediction situations, the threshold for which is also hard-coded). In contrast, GPEP allows users to define the static
and dynamic predictors used for different target variables. GPEP supports the regridding and transformation of dynamic input
data as well.
**Distance-based weight:** GMET calculates local weights for the regression using a hard-coded exponential function based on
the distance between two points, and this choice can have a strong influence on regression estimation. GPEP supports any
user-defined distance functions based on the two parameters: *dist* (distance between points) and *maxdist* (max distance in
weight calculation). This feature facilitates research on the impact of weight functions on regression and ensemble generation
performance.
**Table 1. Comparison of GPEP and GMET methodological features**

|  | GMET v2.0 | GPEP |
|---|---|---|
| Variable | Fixed: precipitation, air temperature, and temperature range | User defined |
| Spatial interpolation | Locally weighted regression<br><br>-    Linear regression | Local regression<br><br>-    Linear regression |



| | | - Logistic regression |
|---|---|---|
| | | - Scikit-learn methods |
| | | Global regression |
| | | - Scikit-learn methods including machine learning methods such as random forest and multi-layer perceptron |
| Prediction uncertainty estimation | - K-fold sample cross-validation for each target grid point | - Cross-validation at station points only, with interpolation to grid points<br><br>- Leave-one-out for local regression<br><br>- K-fold cross-validation for global regression |
| Spatial correlation length | - User defined | - User defined; or<br><br>- Direct estimation from station data |
| Static predictors | Fixed: latitude, longitude, elevation, North-South gradient, West-East gradient | User defined |
| Dynamic predictors | - Same fixed spatial/temporal format for all dynamic variables | - Independent settings for different variables<br><br>- Flexible spatial/temporal formats<br><br>- Allow spatial interpolation and transformation for any variable |
| Distance-based weights | Fixed formulation with weight or non-weight option | User defined formulation |


## 3.2 New technical and usability features in GPEP


GPEP has a different code design compared to GMET, leveraging features of Python to facilitate its implementation,
debugging, and future improvement.





**Environment:** The Fortran-based GMET has certain prerequisites in terms of computation environment, such as the availability of a Fortran compiler and libraries to support NetCDF file standards and linear algebra libraries (e.g., OpenBLAS). GPEP relies on the installation of at least Python 3, along with Python packages including scikit-learn, scipy, xarray, and dask. Whether GMET or GPEP are more accessible for a user will depend on the user's familiarity and facility with Fortran-related or Python-related computational dependencies. In general, both GMET and GPEP are designed with the use of common and/or open-source dependencies. Given the increasing prevalence of Python usage in the Earth Science community, however, we believe that shifting future GMET development to a Python foundation will foster broader engagement by users and developers from more varied computational backgrounds.

**User control:** As is common with all models and software, GMET has a mixture of hard-coded settings or parameters and those that are exposed in configuration files to give the user control over the GMET application. As it has developed, more parameters have been exposed to increase GMET flexibility, and with GPEP we accelerate this trend, either through bringing parameters of interest into the user control file or providing more methodological options. Examples include the spatial correlation length for Tmean and Trange, or Box-cox transformation exponent. The GPEP user can specify (in the configuration file) previously fixed implementation details such as the names of the input dataset dimensions and static predictor variable names (e.g., 'elevation'). Although not strictly necessary for GMET and GPEP operation, these settings allow the user to avoid pre-processing inputs to exacting formats and may enhance the tool's usability.

**Input station data file format:** GMET was coded to read station data timeseries dataset from individual files, along with a single CSV metadata file; whereas GPEP can either use this input file organization, or a single netCDF file that combines all stations and their metadata attributes. The latter approach may be more convenient for users who prefer to bundle the station timeseries into a single file, and the single self-documenting file is faster to read than individual files. It may be less convenient if the station dataset changes frequently (either in the number of stations or length). If used with individual station data files, GPEP will write a merged NetCDF station file to provide the user with both options on subsequent runs.

**Input and output variable specifications:** GMET is currently coded for its most common application -- i.e., reading precipitation and temperature extrema (minimum and maximum) and writing precipitation and temperature mean and range (over the timestep), which are estimated as the mean and difference of the extrema respectively. For many daily meteorological applications, these are the most widely available and used variables. For ensemble member generation, the SCRFs of precipitation and temperature are explicitly linked (via cross-correlation). One of the most important new features of GPEP is to generalize GMET to allow the user to specify arbitrary input and output variables and linkages and transformations between them. In the configuration file, arithmetic expressions can be used to convert input variables to output variables, and the concept of POP is generalized to 'probability of event' (POE), which can be estimated for any variable and can also use a user-defined





event threshold. Users can also define the interdependence of variables in the ensemble generation step directly in the configuration file.

**Neighbouring stations:** GMET allows users to define a fixed number of neighbouring stations used in local regression, while GPEP allows users to define the minimum and maximum numbers of neighbouring stations. This feature responds to the reality that for large domains, users may want to use different numbers of neighbouring stations for areas with different station densities. For example, it may be optimal to use fewer neighbouring stations in remote areas (e.g., northern Canada) to avoid involving stations without notable correlation to the target point, while more neighbouring stations can be used in densely gauged areas (e.g., the eastern U.S.).

**Reproducibility and random field output:** GMET by default uses a random seed when generating ensemble output, whereas GPEP gives users the option to fix (set) the seeds that control the random processes, such as SCRF generation and machine learning initial states. Fixing the random seeds will obtain the same ensemble outcomes from each GPEP run, enabling reproducibility that can be useful in debugging and development. GPEP also provides users with an option to output SCRF values, which may be of interest in development or for certain applications.

**Parallelization:** Computational efficiency is critical for operational application. Python is inherently slower than Fortran for many operations, and GPEP's production of ensemble analyses overall appears to be between 10 and 50 times slower than GMET, based on exploratory benchmarking. For instance, Python is around 10 times slower than Fortran for least-square linear regression functions. For complex computations and loops, the speed gap could be larger. Thus, we have parallelized GPEP's most time-consuming parts using the *multiprocessing* package to improve its speed (future versions may use other packages such as Dask). To demonstrate the parallel efficiency, we tested two locally weighted regression methods (LWR: LWR1 and LWR2) and a global regression method (i.e., RF) for the GMET version 2.0 test case of daily meteorological forcing generation for February 2017 in California, US (Bunn et al, 2022). LWR1 represents the default GMET method using locally weighted linear and logistic regression. LWR2 represents scikit-learn's ridge regression and logistic regression, and RF represents the random forest regressor and classifier. Figure 1 shows that the default LWR1 functions are faster than LWR2, but both methods are slower than the global regression method RF. We observed a significant speedup for LWR1/LWR2 when CPUs increased from 1 to 25 and for RF when CPUs increased from 1 to 15. The speedup for RF diminishes because the compute time is relatively short for lower numbers of CPUs. For generating ensemble members, parallel efficiency remains high with increasing CPU numbers up to 35, as different ensemble members can be generated simultaneously and can fully utilize the available CPUs.

**Table 2. Comparison of GPEP and GMET usability and technical features.**





| | GMET | GPEP |
|---|---|---|
| Environment | Requires a Fortran compiler and associated libraries (e.g., OpenBLAS), and uses standard Fortran compilation approaches. | Requires a Python 3 environment and associated libraries (e.g., Xarray, Dask), and uses standard Python package installation approaches. |
| User settings | - A small number of necessary run settings and parameters are set in the user control files<br><br>- Fixed variable and dimension names for domain and attribute files (do not need to be set) | - A larger number of run settings and parameters are set in the user control files<br><br>- Variable and dimension names are defined in the configuration file (must be set) |
| Input file format | - Individual station data files and a metadata file | - Individual station files and a metadata file; or<br><br>- A combined station file including metadata |
| Variable input and output control | - Probability of precipitation<br><br>- Fixed Prcp-Trange dependence<br><br>- min/max temperature inputs to mean and range of temperature outputs | - Probability of events for any variable<br><br>- Any pair of variables can be linked<br><br>- Arbitrary transformation from input variables to output variables |
| Neighbouring stations | Fixed number defined by users | Min/Max number defined by users |
| Relative speed | Fast | Slow |
| Parallelization | External (accomplished through time-space domain splitting) | Internal (accomplished through multipool processing) |

251



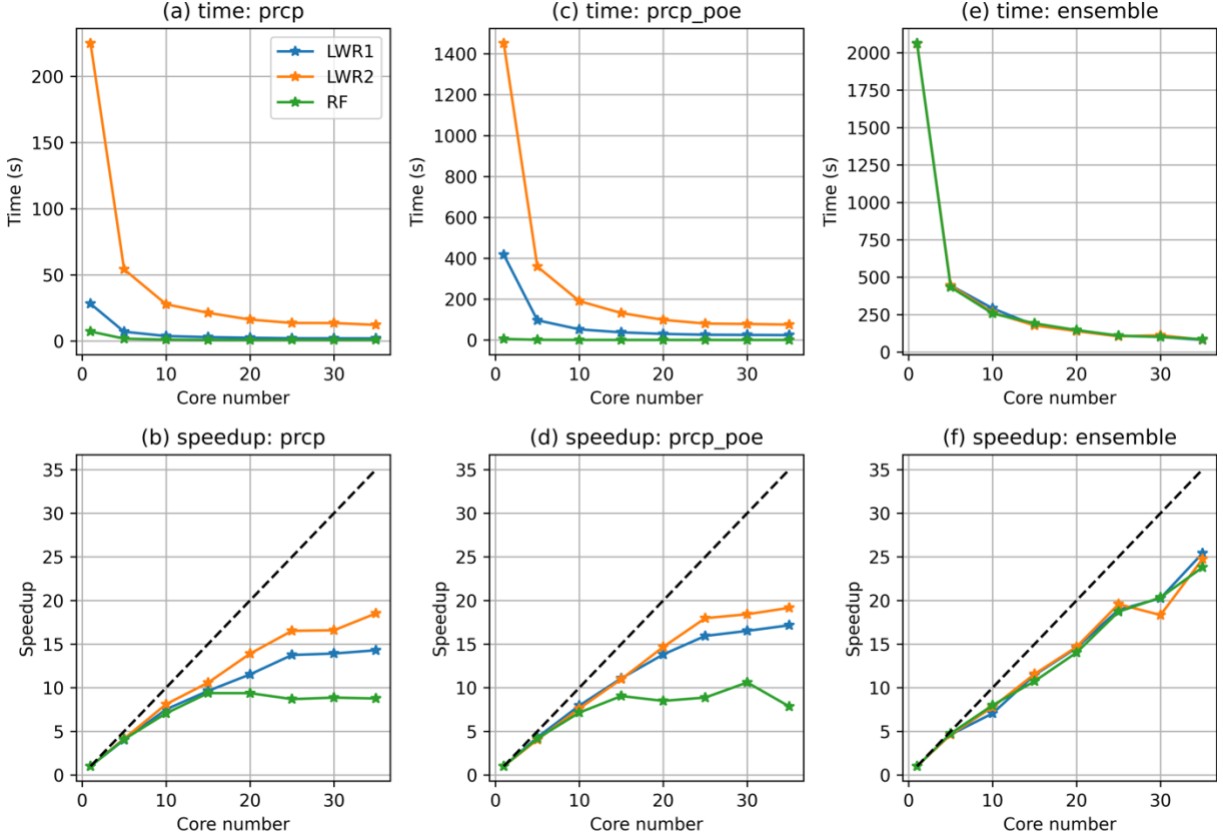

**Figure 1: The CPU-scaling of the time cost (first row) and speed up (second row) of precipitation (prcp) regression (first column), the probability of event for precipitation (prcp_poe) regression (second column), and the generation of 100 ensemble members (third column). Speedup is the ratio between compute time with 1 CPU versus with multiple CPUs.**

**3.3 GPEP documentation**

GPEP comes with extensive documentation that is available on the GitHub repository and provides detailed information on how to set up the environment and how to prepare the configuration file and run GPEP. The documentation includes a comprehensive list of all the available parameters and options that can be used to customize the GPEP input and output.

**4 Demonstration Experiments**

We demonstrate a subset of GPEP capabilities through a small number of experiments described in this section. The first (section 4.1) compares GPEP outcomes to those of GMET for the primary GMET test case, a 1/16th degree resolution daily





meteorological ensemble generation for California, that is included in the GMET version 2.0 repository (Bunn et al, 2021).
The second demonstration (section 4.2) is for meteorological ensembles in a higher resolution (0.01 degree or approximately
1 km) domain including the US Rocky Mountain headwaters of the Colorado headwaters, and the third (section 4.3) illustrates
the use of GPEP to generate ensemble analyses of SWE for the same domain.

**4.1 GMET and GPEP comparison**

In this experiment, we compared the outputs of GPEP and GMET using the GMET version 2.0 test case in California, US.
Figure 2 depicts the agreement between the GMET and GPEP regression model mean estimation of the four primary GMET
output variables, focusing on the locally-weighted linear and logistic regression method based on static predictors only. For
precipitation, Tmean, and Trange, the GPEP and GMET estimates are almost identical for all samples, with the data pairs for
all time steps and grid cells in the domain mainly located along the 1-1 line. For Tmean and Trange, some subtle differences
within $\pm 0.1°$ are observed in the eastern parts of the domain. The differences in the precipitation POE are slightly larger, likely
due to the iterative algorithm of logistic regression amplifying small numerical differences. GPEP tends to generate lower
precipitation POE than GMET for low precipitation probability, while for high POE, GPEP generates higher probabilities. The
positive and negative differences do not show observable spatial patterns. In general, GPEP's mean precipitation POE is
slightly higher than that of GMET by 0.005 (~1%), which is negligible. These results demonstrate that GPEP can reproduce
GMET's grid cell regression estimates with the most common configuration used in GMET applications to date. Note, we do
not compare the ensemble member outputs here. The random fields generated by GMET are challenging to reproduce exactly
in GPEP for a meaningful comparison, and the transformation of the regression models to ensemble members through the
application of SCRFs is a straightforward arithmetic operation.





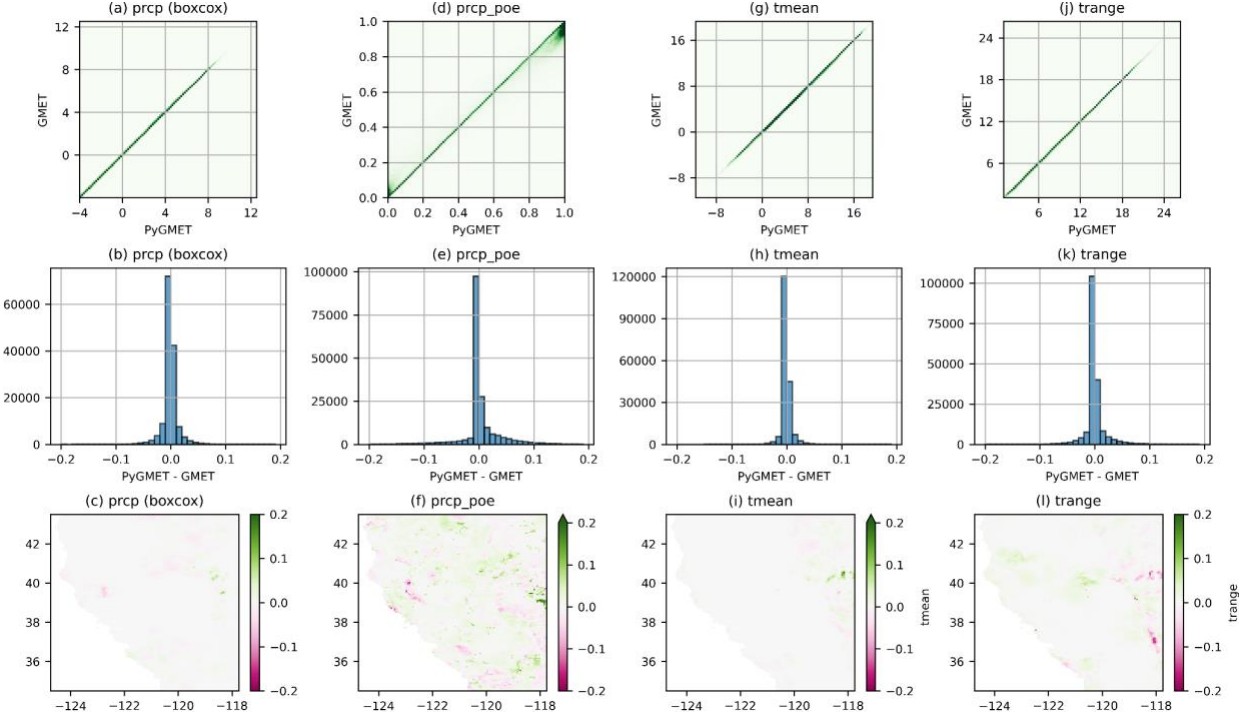

**Figure 2: The scatter density plots (first row) between GPEP and GMET estimates of precipitation (prcp) after Box-cox transformation with a minimum value of -4, precipitation probability of the event (prcp_poe), mean air temperature (tmean) and daily temperature range (trange). The second and third rows show the histograms and spatial distributions of the difference between Python and Fortran outputs. The first and second rows are based on samples from all time steps and grid cells in the domain.**

**4.2 High-resolution meteorological forcing ensemble generation**

Previous GMET-based datasets were all created at mesoscale resolutions, such as 1/16th degree (~6 km) and 0.1° (~10 km). In this experiment, we demonstrate the production of higher resolution ensemble meteorological analyses of daily precipitation, Tmean, and Trange, using a resolution of 1 km in the US upper Colorado region, as shown in Figure 3. The original GMET dataset for this domain was developed for water resources research projects supporting the US Bureau of Reclamation, including a focus on the Colorado Big Thompson Project and hydrologic modeling in the East and Taylor River basins. The elevation ranges between 1427 and 4241 m. The experiment was performed using meteorological data from 864 precipitation and/or temperature stations for the 2013 calendar year. The station observations were quality-controlled (using range and repeating values checks) and filled using a 4-pass iterative quantile mapping from best-correlated nearby stations (Mendoza, et al, 2017; Wood et al, 2023; Liu et al, 2023). Locally weighted linear/logistic regression is used in spatial interpolation. The





static predictors are latitude, longitude, elevation, and south–north and west–east slopes. The slopes are based on smoothed
topography (Figures 3c and 3d) to better characterize orographic precipitation on the windward and leeward sides (Newman
et al., 2015), and the smoothing parameter (a 2-dimensional isotropic Gaussian filter with an effective radius of approximately
100 km) was heuristically selected to maximize the correlation between the slopes and precipitation gradients). In addition,
we use the 2-m air temperature, 2-m dew-point temperature, and precipitation from the ERA5-Land reanalysis product
(Muñoz-Sabater et al., 2021) as dynamic (time-varying) predictors because of their linkage with temperature, humidity, and
precipitation. The static and dynamic predictor selection was for demonstration purposes and does not presume to offer optimal
performance. In practice, users may choose to test different combinations to achieve the best accuracy, which can be
determined through examining cross-validation results.

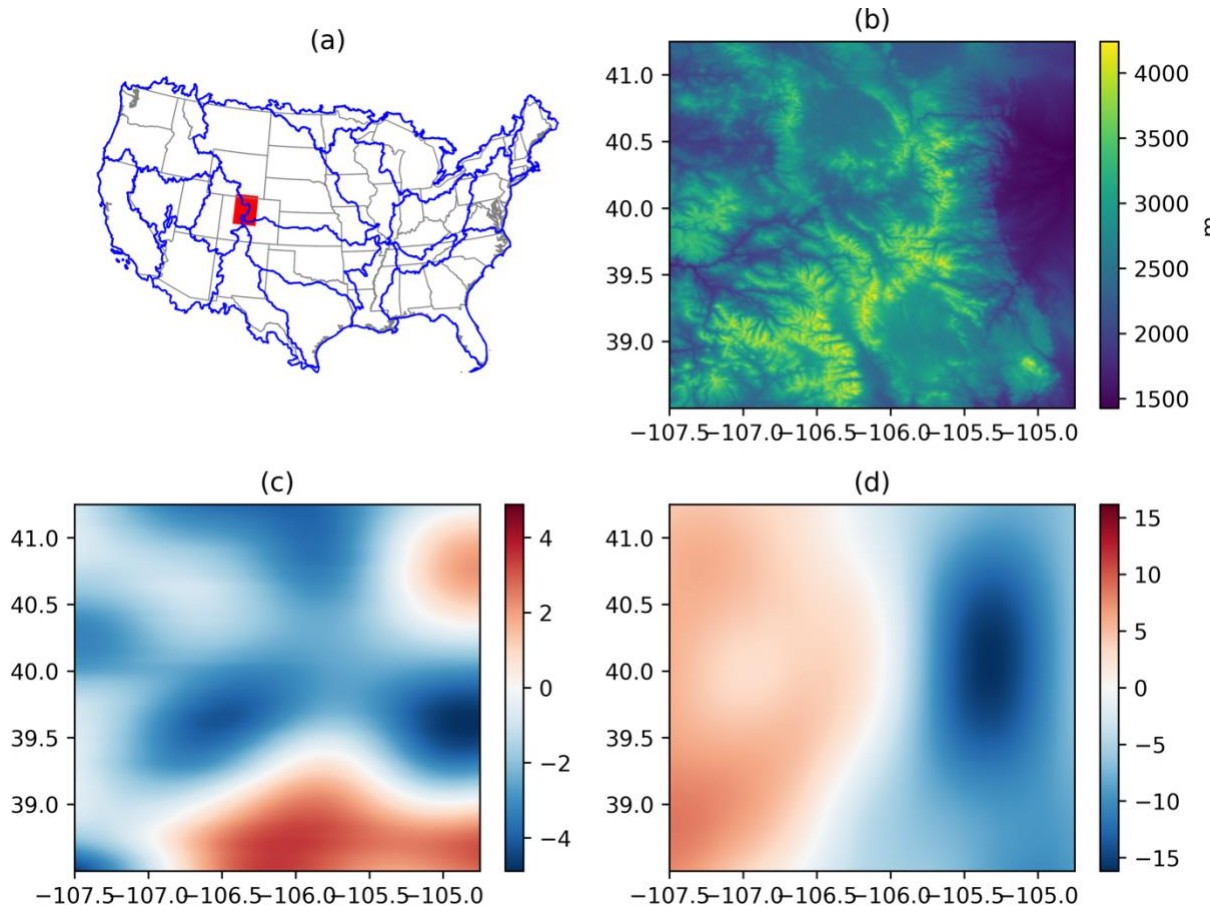






**Figure 3: (a) The location of the test case area in the upper Colorado region, US (red region). Blue lines outline the Hydrologic Unit Code (HUC) level-2 regions. (b) The elevation. (c) and (d) are the south–north and west–east slopes, respectively.**

As introduced in Section 3, GPEP uses the leave-one-out strategy to estimate the uncertainty of local regression. GPEP also provides 16 evaluation metrics in the output file, facilitating assessing the quality of interpolation estimates. For example, Figure 4 displays three metrics, namely, the correlation coefficients (CC: $0-1$), mean absolute error (MAE: $0-\infty$), and the modified Kling-Gupta efficiency (KGE": $-\infty-1$). KGE" (Tang et al., 2021) uses the standard deviation instead of the mean value to normalize the bias term, making it suitable for temperature variables. Precipitation estimates show higher accuracy in the relatively flat eastern areas, exhibiting high CC and KGE" and low MAE, while the vast western areas have lower accuracy due to complex terrain. Tmean and Trange exhibit different spatial patterns, with Tmean having much better MAE and KGE" than Trange. This indicates the difficulty in capturing diurnal fluctuations between the minimum and maximum temperature.

Figure 5 shows the spatial distributions of precipitation, Tmean, and Trange from three ensemble members during the period September 9 to 17, 2013, when heavy precipitation occurred with the accumulated amounts exceeding 500 mm at the precipitation center. The large differences between members at event centers originate from the interpolation uncertainties which are mainly caused by the degraded capability of the station network and interpolation method to capture extreme events. The magnitude is generally comparable to other post-flood analyses (e.g., Gochis et al., 2015). Tmean shows the lowest ensemble spread among the three variables, and Trange shows the intermediate ensemble spread.

Figure 6 shows the time series of ensemble outputs in September 2013 for Boulder County, Colorado, parts of which experienced significant extreme precipitation, causing devastating floods from September 11 to 15, 2013. The return periods of the floods were estimated to be 25 to 100 years. The GPEP ensemble precipitation indicates a major precipitation event (Figure 6a) with mean or median precipitation going beyond 60 mm/d and some members going beyond 100 mm/d around September 11. For precipitation estimation, it is possible that the use of a wind speed and direction dynamic predictor would also contribute to an upslope precipitation enhancement, leading to higher intensities at elevation in the Front Range basins that experienced flooding. The flooding period also suffers from the largest uncertainty in September with the 5%-95% bounds ranging between <10 mm/day and >150 mm/day. This illustration highlights the challenge of accurately capturing extreme events with deterministic precipitation estimation and the potential usefulness of ensemble estimation in representing uncertainty and triggering useful alerts for extreme events with their upper bounds. Additionally, Tmean displays a decreasing trend accompanied by continuous precipitation, while Trange shows an inverse trend to Tmean after September 8.

337

**Figure 4: The spatial distributions of CC (first row), MAE (second row), and KGE'' (third row) for precipitation (first**
**column), Tmean (second column), and Trange (third column) based on leave-one-out validation.**





**Figure 5: The spatial distribution of total precipitation and mean Tmean/Trange (columns) from three ensemble members (rows) from September 9 to 17, 2013.**







**Figure 6: The time series of GPEP ensemble outputs in Boulder County, Colorado (39.91° to 40.26°N, -105.7° to -105.05°W).**





### 4.3 Snow water equivalent (SWE) estimation


GPEP can be applied to a wide range of geophysical variables beyond precipitation and temperature, which has been the
common application of GMET. In this test case, snow water equivalent (SWE) is chosen as an example, as it was one of the
first applications of the locally-weighted terrain regression and ensemble generation methodology that was later developed
into GMET (Slater & Clark, 2006). We use the same domain as in the previous test case, and a configuration sharing some
details: the predictors are latitude, longitude, elevation, south–north and west–east slopes, the transformation method was Box-
cox, and the locally weighted linear/logistic regression is adopted. In practice, other predictors such as other topographic
variables, vegetation types, and dynamic predictors such as radiation, temperature, and SWE from models can be explored for
improved performance. We estimate SWE ensembles for the water year from October 2012 to September 2013. The station
observations come from the SNOwpack TELemetry Network (SNOTEL) network. Only serially complete stations (71) in the
study period are used, as we did not attempt to quality control and fill the station data for this demonstration.
Figure 7 shows the LOO cross-validation results of SWE. According to station observations, the SWE peak occurs on April
25, 2013, during the 2012–2013 water year. Overall, the spatial distributions of observed and estimated SWE are similar
(Figures 7a,b). However, the estimated SWE is smoother in space, leading to large biases at a few points. For example, SWE
is overestimated at two stations (~ 39.3°N / 106.6°W and ~ 40.2°N / 105.6°W) that show notably lower SWE than surrounding
stations. For the mean annual SWE (Figure 7c), estimates agree well with observations (the relative mean error for the points
shown is 2.94%), except for one outlier corresponding to the station at 40.35°N / 106.38°W. The station has an elevation of
3340 m, where the estimated SWE is 375 mm but the observed SWE is 180 mm. It is possible that the predictors used in this
demonstration do not represent the factors affecting SWE distribution well, leading to sub-optimal regression results. Figure
7d shows that the seasonal variability of cross-validated GPEP SWE (averaged across the 71 points) in the upper Colorado
region is well captured, except for the underestimation of SWE at the end of the melt period (June 2013). Optimizing this SWE
analysis is beyond the purposes of this capability demonstration, and it is likely that different predictor or methodological
choices would improve the results shown here.
SWE and other hydrologic or land surface variables can be strongly auto-correlated, distinguishing their probabilistic
estimation from most meteorological fields, e.g., precipitation or temperature. The lag-1 auto-correlation of SWE exceeds 0.99
within the study area, implying that the random field in all time steps will be quite similar to that in the first time step (Equation
(1), and the ensemble spread may be underestimated. This example highlights the importance of generating a realistic initial
spatial random field, which significantly depends on the spatial correlation length, for the perturbation of SWE, as well as
predictors that represent factors leading to high-frequency space/time variability in SWE. For demonstration purposes, we
have used a spatial correlation length of 10 km, but would encourage future studies to investigate optimal settings for this
length. Figure 8 illustrates the 25-member SWE estimates. The uncertainty is lower during the accumulation stage and greater





when SWE reaches its peak and melting begins (Figure 8a). Figures 8b and 8c display the ensemble mean and spread of SWE
on April 25, 2013, respectively. Substantial SWE is observed in high-altitude areas, where the spread is also large. Probabilistic
SWE estimates can support the uncertainty quantification of a variety of applications related to water resources management
such as forecasting streamflow, including seasonal runoff volumes for managing reservoirs and assessing flood risks.

**Figure 7: (a) SWE of station observations on April 25, 2013, when the mean SWE reaches the peak, (b) SWE of leave-**
**one-out interpolation estimates on April 25, 2013, (c) scatter plots between observed and estimated mean annual SWE**
**with the colour representing KGE", and (d) the variability of daily SWE.**



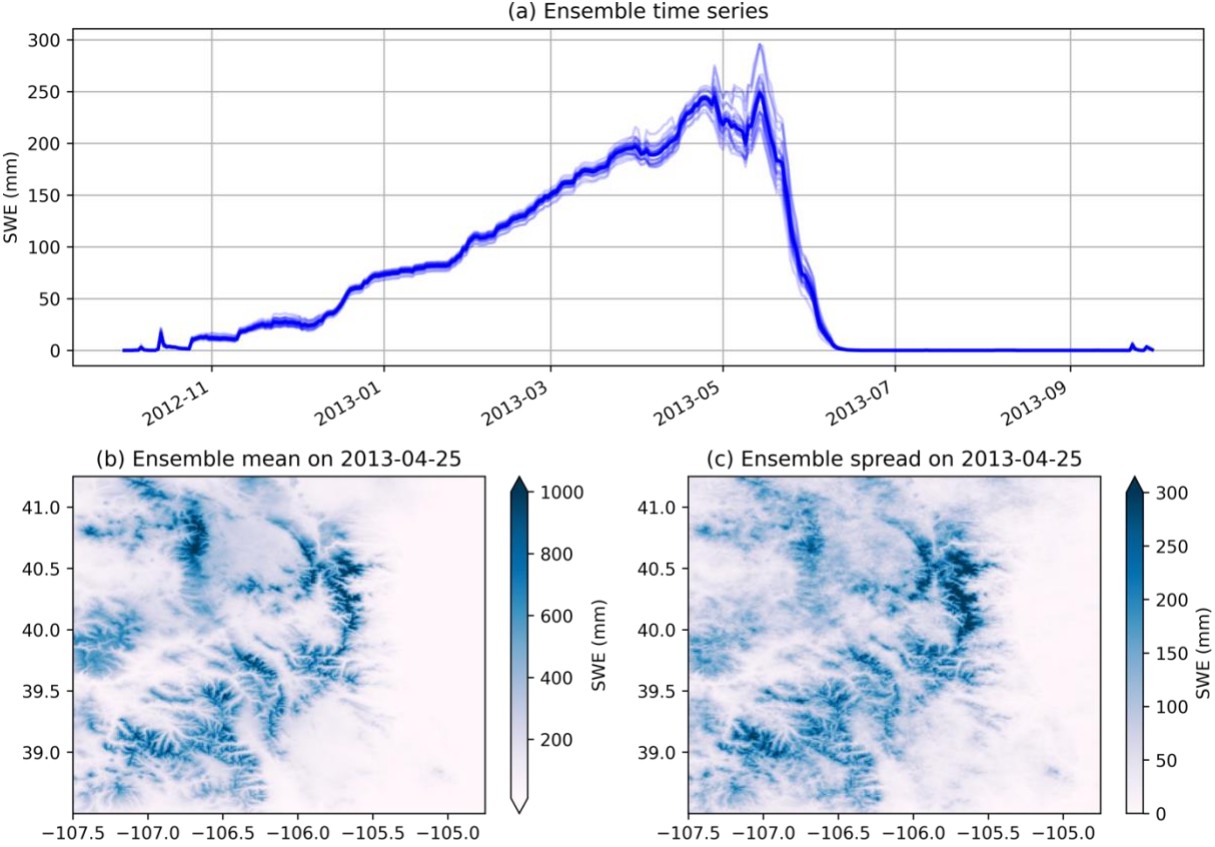

**Figure 8: (a) Mean daily SWE in the study area from 25 members. The dark blue line is the ensemble mean. (b) and (c) are ensemble mean and ensemble spread of SWE on April 25, 2013, respectively.**

**5 Summary and discussion**

GPEP is a powerful Python-based software for ensemble, probabilistic estimation of any geophysical variable. It expands on the capabilities offered by the Fortran-based GMET software, on which GPEP is based. GPEP supports various local and global regression methods including ML techniques for spatial interpolation and fusion of multi-sensor datasets, and can generate any number of ensemble members using the predictive uncertainty results obtained from cross-validation. Although GPEP is far slower than the original GMET, the tool's internal parallelization capability scales well to improve its computation efficiency, making it suitable for both research and operational applications. The Fortran-based GMET has been used for almost a decade in numerous hydrology and water resources applications, demonstrating its quality and value through the performance of GMET datasets relative to other widely used options. The central motivations for translating GMET to Python were to broaden the development community for the probabilistic estimation tool, and to facilitate more rapid development





with linkages to ML methods through the growing Python-based activities and resources in this area. The capabilities of GPEP
that extend GMET illustrate this latter potential. Of course, GPEP has much room for improvement. For example, myriad
methodological options exist for variable transformation (the current Box-cox transformation may not be ideal) and can be
added in the future to address the requirement of specific variables (Papalexiou, 2018). Similarly, the generation of
spatiotemporally correlated multi-variable analyses can benefit from the addition of a variety of methods, including Papalexiou
& Serinaldi (2020) technique to construct flexible spatiotemporal correlation structures by combining copulas and survival
functions, and geostatistical tools such as the Python-based GSTools (Müller et al., 2022) that can be used to generate spatial
random fields. The current sklearn method libraries are also just a starting point for expanding the options available for
conditional estimation of geophysical fields, and we expect that future development may link to ML and deep learning
packages such as PyTorch, TensorFlow, or Keras, as the field evolves. By incorporating these and other potential options,
GPEP can become even more versatile in hydrometeorology and Earth Science studies. Finally, the single largest drawback to
the move from the Fortran-based GMET to GPEP is the significantly slower outcomes for current meteorological GMET
applications. Work to understand and optimize this aspect has only begun, posing challenges for GPEP in its current large-
domain and near-real-time operational application. We are optimistic that this issue can be resolved through further efforts to
optimize the algorithms, hybrid programming for the time-consuming part of GPEP, and explore additional parallel processing
options, and shift development from CPU-based computing toward using GPUs. This paper documents the initial
implementation of GPEP with the aim of attracting a community of collaborators who will help to achieve some of these future
developments.

*Code and data availability.* GPEP is available at GitHub (https://github.com/NCAR/GPEP) and Zenodo
(https://doi.org/10.5281/zenodo.8223175). The California precipitation/temperature and Upper Colorado SWE test cases are
available at https://zenodo.org/record/8222852.
*Author contributions.* GT refactored and expanded GMET into GPEP, and GT wrote the first draft of the paper and produced
all paper analyses, with guidance from AW. AW co-wrote the final paper, contributed the test case datasets, and helped to
guide the design, usability, and testing of GPEP. GPEP development was funded by a USACE project at NCAR led by AW,
and also drew on pieces of code written by GT at the U. of Saskatchewan. AN, MC, and SP provided comments and edits on
the final paper draft.
*Competing interests.* The authors declare to have no competing interests.



*Acknowledgements.* This study is supported by the research grants to NCAR from the United States Army Corps of Engineers Climate Preparedness and Resilience Program and the United States Bureau of Reclamation Science and Technology Program. We acknowledge high-performance computing support provided by NCAR's Computational and Information Systems Laboratory, sponsored by the National Science Foundation.

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
