# Peer review of "GPEP v1.0: a Geospatial Probabilistic Estimation Package to support"

_Geoscientific Model Development, 2023_

## Referee Comment (RC1)

**General Comments**

Tang et al described a new developed geospatial probabilistic estimation package (GPEP) to generate ensemble meteorological datasets based on station observations. They demonstrated good performance of GPEP with some examples. Compared to Gridded Meteorological Ensemble Tool (GMET), GPEP has several improvements, such as multiple selection of spatial interpolation methods, additional meteorological variables, user defined inputs, and etc. However, the computational performance of GPEP can be worse than GMET since GPEP is developed in Python. The computational performance is improved by using multiprocessing package, and they show speed of the simulation is significantly increased as more cores are used. GPEP represents a useful tool for the Earth science community. Specifically, the gridded meteorological variables that generated by GPEP can be used as forcing data for Earth System Models, hydrological models, especially for ungagged regions. The ensemble estimations can be further used for assessing the uncertainty caused by meteorological forcings in the simulation. Although I think this study could be a significant contribution to ensemble geophysical datasets estimation, it can be further improved before publication in Geoscientific Model Development. Please find my major concerns and specific comments in the following.

**Major Comments**

1. GMEP considers the spatial correlation and correlation between temperature and precipitation. As more variables can be processed and generated in GPEP, are the intercorrelation among variables considered? For example, in the application of Earth System Model, precipitation, humidity, radiation, and temperature are needed. Those variables are correlated to each other in time and space. Is it possible for GPEP to generate those variables together? In addition, how other variables are generated in GPEP is not described. Eq (1) – Eq (4) describes how temperature and precipitation are generated in GMEP. Can the same equations be used for any other variables?

2. What is the roadblock for running GMEP for the High-resolution meteorological forcing ensemble generation? It will be interesting to see if GPEP has better or similar performance than GMEP in such application.

3. There are other 1km reanalysis meteorological datasets that can be used for benchmarking GPEP's simulation for the high-resolution meteorological forcing ensemble generation example (i.e., Sec 4.2). For example, Daily Surface Weather Data on a 1-km Grid for North America (Daymet). Is GPEP better than existing high-resolution reanalysis dataset for reproducing the meteorological variables at the station locations? How is the spatial pattern compared to such high-resolution reanalysis dataset. I think by adding such benchmark and evaluation can give the readers/users more confidence on the application of GPEP.

4. The authors argued GPEP features multiple selection of spatial interpolation method. But still the locally weighted linear/logistic regression were used in all the demonstrations. I think it is necessary for the author to show the application of other spatial interpolation method. Specifically, will using supervised learning method for the spatial interpolation can

improve the performance in Figure 4 (the high-resolution meteorological forcing ensemble generation)?

5. Does GPEP only accept gauge data as input? Can we use gridded reanalysis dataset as input and output at higher spatial resolution? In addition, can GPEP generate the meteorological datasets at sub-daily scale? This can be useful for using GPEP to generate forcings for models, as some models requires sub-daily meteorological forcings.

**Specific Comments**

Line 32: In my experience, reanalysis dataset commonly has better temporal coverage than the station observation. Please cite relevant reference to support this statement.

Line 46: Full name for HadCRUT4?

Line 88 and Line 89: What are continental EMDNA and global EM-Earth?

Line 98: Is temperature range = maximum temperature – minimum temperature in the day?

Line 235: I wonder the computational performance for large scale application, for example, continental or global simulation.

Section 4.1: I suppose same algorithm was adopted in GPEP. So, is the difference caused by random seed used in GMET? Can the authors further explain the attribution for the differences?

Figure 2: Does subplot (a) represent the mean precipitation of the simulation period, or all the daily precipitations within the simulation period?

Figure 3: What is the source and spatial resolution of the elevation data? Does GPEP estimate the south-north and west -east slopes internally, or the user need to preprocess it? It will be useful for the authors to describe how the slopes were calculated.

Line 317: Could the higher performance in the flat eastern areas due to that there are more stations in this region? Then this raise the question what is the minimum number of stations that needed by GPEP for a good performance? What is the performance of GPEP in data sparse region? I am not asking the authors to run additional simulations for this question, some discussions and perspective from the authors will be very helpful for the readers.

Line 322 – line 325: Can the authors quantify the ensemble spread? For example, the spatial correlation between any two ensemble members.

Line 326: Is there a station fall inside the area selected by Figure 6? If so, how the simulation ensemble compared to observation?

---

## Author Comment (AC1)

**General Comments**

Tang et al described a new developed geospatial probabilistic estimation package (GPEP) to generate ensemble meteorological datasets based on station observations. They demonstrated good performance of GPEP with some examples. Compared to Gridded Meteorological Ensemble Tool (GMET), GPEP has several improvements, such as multiple selection of spatial interpolation methods, additional meteorological variables, user defined inputs, and etc. However, the computational performance of GPEP can be worse than GMET since GPEP is developed in Python. The computational performance is improved by using multiprocessing package, and they show speed of the simulation is significantly increased as more cores are used. GPEP represents a useful tool for the Earth science community. Specifically, the gridded meteorological variables that generated by GPEP can be used as forcing data for Earth System Models, hydrological models, especially for ungagged regions. The ensemble estimations can be further used for assessing the uncertainty caused by meteorological forcings in the simulation. Although I think this study could be a significant contribution to ensemble geophysical datasets estimation, it can be further improved before publication in Geoscientific Model Development. Please find my major concerns and specific comments in the following.

Response: Thank you for the comprehensive review and insightful feedback on our manuscript. We have addressed each of your comments in detail below and revised the manuscript accordingly.

**Major comments:**

1. GMEP considers the spatial correlation and correlation between temperature and precipitation. As more variables can be processed and generated in GPEP, are the intercorrelation among variables considered? For example, in the application of Earth System Model, precipitation, humidity, radiation, and temperature are needed. Those variables are correlated to each other in time and space. Is it possible for GPEP to generate those variables together? In addition, how other variables are generated in GPEP is not described. Eq (1) – Eq (4) describes how temperature and precipitation are generated in GMEP. Can the same equations be used for any other variables?

Response: We believe you are referring to GMET when you mention GMEP. The problem you point out is valuable. In the original GMET methodology, the dependence relationship between precipitation and daily temperature range is defined via their cross correlation, as depicted in Equation (1). GPEP extends this capability, allowing users to specify dependencies between any pair of variables, such as precipitation versus temperature range, or precipitation versus humidity, or temperature versus radiation, through the "linkvar" parameter in the configuration file.

However, the current implementation does not handle intercorrelation relationships among more than two variables. For example, for precipitation, humidity, radiation, and temperature, Equation (1) cannot consider how each variable is correlated to the other variables. A workaround is to decompose the intercorrelation into multiple binary dependencies. For example, we can define three dependence relationships here: (1) humidity depends on temperature, (2) precipitation depends on temperature, and (3) temperature depends on radiation. GPEP can address this by first

generating radiation random fields, followed by temperature random fields based on the radiation, and subsequently, humidity and precipitation random fields based on the temperature. However, this solution is less efficient than jointly estimating multivariate cross-correlation or covariation matrices as part of a generalized multiple linear regression approach.

Generalizing the current bivariate cross-correlation implementation is a high priority for future development, but will require some further code development effort. We are also interested in exploring alternatives such as using copula functions, which may be facilitated by providing a python code-base which links to more existing packages than Fortran. In any case, we have added a new comment in the Discussion section of revised manuscript recognizing this current limitation as a future research direction.

About your second question, the answer is that other variables can also be generated using Eq (1) – Eq (4). The probabilistic estimation framework is applicable to any variable following a normal distribution. For variables that deviate from this distribution, such as precipitation or snow water equivalent, a transformation is necessitated, as well as (depending on application) the treatment of the variable as intermittent, with an associated probability of event. GPEP users have the flexibility to adjust the exponential factor of the box-cox transformation in the configuration file to ensure the transformation is suitable for their target variables, and in the future will have a broader range of transformation options. We have explained this in Section 2.

2. What is the roadblock for running GMEP for the High-resolution meteorological forcing ensemble generation? It will be interesting to see if GPEP has better or similar performance than GMEP in such application.

Response: This question is partly answered in Section 4.1, which compares GMET and GPEP in a 1/16th degree resolution daily meteorological ensemble generation for California. While the term "high resolution" might be subjective and vary among individuals, the mesocale 1/16th degree results indicate that when GPEP is configured to mirror GMET settings, both tools can yield nearly identical estimates.

Roadblocks for large-domain high-resolution estimation currently the computational efficiency of Python, although parallel computation can alleviate this problem (Figure 2). Other options include algorithm optimization, hybrid programming, and shift development from CPU-based computing toward using GPUs. We have added discussion in Section 5. Another issue that may be particular to applications in which the predictors are highly variable at granular, high resolution scales, is that they can require a greater degree of extrapolation, which may give rise to unrealistic behavior at extreme locations from a predictor or predictand perspective. For example, a terrain-based predictor at fine resolutions can have a much larger range (spanning elevations perhaps 1000-5000m) while the training data is unlikely to fully represent observations across this full range. Co-author Wood has run GMET experimentally at resolutions up to 250m, and noted some issues in this regard. Solutions may include different forms of regression (such at gaussian process regression) that recognize variable data support, or certain machine learning approaches, and adjust the uncertainty estimation accordingly.

3. There are other 1km reanalysis meteorological datasets that can be used for benchmarking GPEP's simulation for the high-resolution meteorological forcing ensemble generation example (i.e., Sec 4.2). For example, Daily Surface Weather Data on a 1-km Grid for North America (Daymet). Is GPEP better than existing high-resolution reanalysis dataset for reproducing the meteorological variables at the station locations? How is the spatial pattern compared to such high-resolution reanalysis dataset. I think by adding such benchmark and evaluation can give the readers/users more confidence on the application of GPEP.

Response: A short answer is that we haven't had the scope of effort to include a detailed Daymet (or other dataset comparison), although a new study by a PhD student of Co-author Wood intercomparing GMET/GPEP with 5-6 other dataset options is underway. It is an excellent topic, but not one we can delve deeply into here. We agree that adding a high-resolution dataset as the benchmark would be an interesting comparison point for readers, although we do not know the absolute quality of Daymet, and the GPEP case study is not a production quality dataset. There are several reasons that we do not include the comparison in the manuscript.

(1) A prior paper by Henn et al. (2018) assessed the differences in gridded precipitation datasets in complex terrain (e.g., the western CONUS), including the GMET-based CONUS dataset (Newman et al., 2015), Daymet, and some other widely used datasets. In this manuscript, we have demonstrated that GPEP can reproduce the outputs of GMET with the same settings used in Newman et al. (2015). We have touched on the comparison results from Henn et al. (2018) in the revised manuscript.

(2) GPEP is a tool with myriad configuration choices for estimation, rather than a program with a fixed setup to produce only meteorological datasets. Numerous datasets can be generated by changing parameters in the configuration file, while exploring optimal configurations or comparing a specific configuration to existing datasets has gone beyond the scope of this software paper. Users need to fine-tune GPEP's settings to achieve optimal performance tailored to specific regions or variables, and we have mainly provided case studies to illustrate the working of GMET without tailoring them as standalone, high-quality datasets for broader use (like Daymet).

(3) Cross-validation capabilities: A notable feature of GPEP is its ability to offer cross-validation results, as illustrated in the manuscript. This allows users to gain an objective insight into GPEP's statistical accuracy or reliability without comparing it to other datasets – essentially, it quantifies its own ability to estimate out of sample observations. Evaluating through comparison to existing datasets like Daymet can be challenging since they often integrate ground station data into their production.

We have added discussions in Section 4.1 and Section 5.

Reference:

Henn, B., Newman, A. J., Livneh, B., Daly, C., & Lundquist, J. D. (2018). An assessment of differences in gridded precipitation datasets in complex terrain. Journal of Hydrology, 556, 1205–1219. https://doi.org/10.1016/j.jhydrol.2017.03.008

Newman, A. J., Clark, M. P., Craig, J., Nijssen, B., Wood, A., Gutmann, E., et al. (2015). Gridded Ensemble Precipitation and Temperature Estimates for the Contiguous United States. Journal of Hydrometeorology, 16(6), 2481–2500. https://doi.org/10.1175/JHM-D-15-0026.1

4. The authors argued GPEP features multiple selection of spatial interpolation method. But still the locally weighted linear/logistic regression were used in all the demonstrations. I think it is necessary for the author to show the application of other spatial interpolation method. Specifically, will using supervised learning method for the spatial interpolation can improve the performance in Figure 4 (the high-resolution meteorological forcing ensemble generation)?

Response: Thanks for the suggestion, and we agree that it is important to demonstrate machine learning-based experiments in the manuscript. In Figure 2, we have shown the comparison of the parallel efficiency between locally weighted regression and random forest. In light of your feedback, we have further enriched our manuscript by incorporating experiments that utilize supervised learning techniques and compare machine learning's performance with that of the locally weighted regression in Figure 5 (formerly Figure 4).

We added a new figure (i.e., Figure 6) in Section 4.2 to compare the performance of random forest estimates and locally weighted regression. However, please note that this just demonstrates an application of one machine learning method. Exploring the superiority of machine learning versus locally weighted regression would necessitate a comprehensive exploration of hyperparameters and feature combinations to optimize machine learning performance. Conclusions may also vary for different regions. The GPEP software or the manuscript cannot provide a more quantitative and comprehensive conclusion here.

5. Does GPEP only accept gauge data as input? Can we use gridded reanalysis dataset as input and output at higher spatial resolution? In addition, can GPEP generate the meteorological datasets at sub-daily scale? This can be useful for using GPEP to generate forcings for models, as some models requires sub-daily meteorological forcings.

Response: This is definitely possible with GPEP (and GMET v2.0, albeit with more pre-processing required for the latter). A major goal of allowing for dynamic predictors in both software was to enable a fusion of station and other forms of information, such as gridded analyses (reanalysis, NWP, satellite, radar). Hence GPEP can work with station data and also incorporate reanalysis datasets as dynamic predictors for spatial estimation. If users wish to use reanalysis data as direct inputs, they can reformat these datasets by treating each grid cell as an individual station. This involves reshaping the reanalysis grids from a (time, x, y) structure to a (time, x·y) format. Once reformatted, these can be fed into GPEP as standard inputs, allowing users to generate outputs at their desired spatial resolution.

About the second question, GPEP can generate forcings on any temporal scale, including sub-daily intervals as long as sub-daily input data are fed into the package.

We have explained more about the two questions in Section 5.

**Specific Comments**

Line 32: In my experience, reanalysis dataset commonly has better temporal coverage than the station observation. Please cite relevant reference to support this statement.

Response: We intended to convey that ground stations offer the longest meteorological observations, while reanalysis datasets are estimates. To avoid confusion, we have removed the word "longest".

Line 46: Full name for HadCRUT4?

Response: We have included the full name, i.e., Hadley Centre/Climate Research Unit Temperature version 4.

Line 88 and Line 89: What are continental EMDNA and global EM-Earth?

Response: The full names and references of the two datasets were mentioned in Line 49 and Line 50.

Line 98: Is temperature range = maximum temperature – minimum temperature in the day?

Response: Yes, the temperature range is defined as the difference between the maximum and minimum temperatures of the day. We have clarified is in the revised manuscript.

Line 235: I wonder the computational performance for large scale application, for example, continental or global simulation.

Response: The speed of GPEP is influenced by many factors, such as regression parameters, chosen resolution, and domain size. Our upper Colorado test case, with a 0.01-degree resolution, has 275×275 grids. In comparison, the North American Land Data Assimilation System (NLDAS) uses a 0.125-degree resolution, resulting in 224×464 grids. The upper Colorado case, having about 73% of NLDAS's grid count, can already provide a benchmark for larger domain applications. In the upper Colorado test case, we utilized 36 CPUs on the Casper HPC at NCAR. It took 5,560 seconds to produce regression estimates and 36 ensemble members for the year 2013. Note that this duration does not account for the one-time generation of prior files, such as indices for neighboring stations and the spatial correlation structure.

Additionally, while GPEP supports locally weighted regression, it also offers faster global machine learning regression methods. Direct comparisons with the GMET package can be challenging due to methodological differences.

We have added some further explanations for Figure 2 and in Section 4.2 so users can have a basic estimate of the computation time for a similar application.

Section 4.1: I suppose the same algorithm was adopted in GPEP. So, is the difference caused by the random seed used in GMET? Can the authors further explain the attribution for the differences?

Response: Indeed, both GPEP and GMET employ the same locally weighted regression algorithm. The differences observed aren't due to the random seed, as Section 4.1 focuses on deterministic estimates. While GPEP is Python-based and GMET is Fortran-based, their algorithmic performance is consistent. Differences in their cross-validation approaches can also lead to differences in their deterministic and uncertainty estimation. We have conducted independent numerical comparisons (i.e., extracting regression functions from GPEP and GMET packages) to confirm this. In Figure 3 (formerly Figure 2), most estimates from GPEP and GMET are the same. But minor discrepancies, especially in the probability of precipitation, also come from slight numerical differences in data inputs, attributed to differences in double precision or single precision in GPEP and GMET codes. These minor variations can be magnified during iterative processes of logistic regression. We have elaborated on this in our revised manuscript.

Figure 2: Does subplot (a) represent the mean precipitation of the simulation period, or all the daily precipitations within the simulation period?

Response: It represents all the daily precipitations within the simulation period. Specifically, each point corresponds to the values for a single grid at a particular time step. We have clarified this in the figure caption.

Figure 3: What is the source and spatial resolution of the elevation data? Does GPEP estimate the south-north and west-east slopes internally, or does the user need to preprocess it? It will be useful for the authors to describe how the slopes were calculated.

Response: The elevation data is from SRTM with an original resolution of 3 arc-seconds. This data was subsequently resampled to a 0.01-degree resolution for the test case. GPEP does not automatically compute the south-north and west-east slopes. Instead, users are responsible for determining the predictors they wish to use in their studies. For the test case presented, the south-north and west-east slopes were derived from a smoothed Digital Elevation Model (DEM) to minimize high-resolution noise and emphasize broader topographic influences. Further details on the slope data preparation have been provided in the revised manuscript. The prior GMET applications have led to the development of a range of grid processing scripts (unpublished so far) to prepare such inputs.

Line 317: Could the higher performance in the flat eastern areas be due to there being more stations in this region? This raises the question: what is the minimum number of stations needed by GPEP for good performance? What is the performance of GPEP in data-sparse regions? I am not asking the authors to run additional simulations for this question; some discussions and perspective from the authors will be very helpful for the readers.

Response: Indeed, the higher density of stations in the flat eastern areas likely contributes to the enhanced performance observed in that region. We have elaborated on this in the revised manuscript.

The question of the minimum number of stations required for GPEP to function optimally is multifaceted. The optimal station density can vary based on environmental conditions, targeted

meteorological variables, and methodology choices. For instance, certain machine learning techniques could be less sensitive to station densities compared to traditional interpolation methods. We have discussed more about this issue in the revised manuscript.

Regarding the performance of GPEP in data-sparse regions, it is hard to predict because of the abovementioned reasons. However, the flexible configurations in GPEP offer users the chance to explore optimal regional performance compared to using public datasets or tools with limited choices. We expect that like all statistical modeling problems, GPEP performance will be undermined by data sparsity, and this was another reason for including the ability to use dynamic gridded predictors, such as reanalysis or NWP – in regions of sparse in situ data, one would do no worse than the gridded background fields, ideally.

We have added discussions in Section 5.

Line 322 – Line 325: Can the authors quantify the ensemble spread? For example, the spatial correlation between any two ensemble members.

Response: To quantify the ensemble spread, one common approach is to use the standard deviation. We have incorporated ensemble spread maps in Figure 7 to provide a clearer visualization of this spread. The ensemble spread of temperature shows that during that period, the uncertainty mainly occurs at stations located in the southern part of the study area.

Line 326: Is there a station located inside the area selected by Figure 6? If so, how does the simulation ensemble compare to the observation?

Response: Yes, some stations are situated within the area highlighted in Figure 7 (formerly Figure 6). We did compare ensemble observations to station observations. The ensemble mean generally agrees well with station observations, which is as expected because the comparison is not independent due to the integration of station data in ensemble estimates. According to the probabilistic method, the probabilistic estimates should fluctuate around the true value if the deterministic regression is accurate enough.

---

## Author Comment (AC2)

The study presents a new open-source Python library for ensemble estimation of geospatial earth system variables. The new library (GPEP) is based on an existing one (GMET) that is programmed in Fortran. The authors aim to increase the flexibility that GMET provides, increasing the amount of variables that can be analyzed, the number of spatial interpolation schemes and other important characteristics. They apply the library to three demonstration experiments where they compare their results to those provided by GMET. They conclude by remarking the advantages of the new library and some of its drawbacks.

The main contribution of this work is to have translated a model in FORTRAN to Python. FORTRAN is an old-school programming language, fast and expressive, that has been extensively used for numerical model programming and other intensive tasks. It is less popular than it used to be and compiling it may be complicated. On the other hand, Python is an interpreted language that does not require to compile its codes, is very portable and with a plethora of libraries around it that automatically creates synergies with every new library, like it could be the case with GPEP. Indeed, the authors mention the possibility to interoperate GPEP with Scikit-Learn, a Python-based machine learning library, which may give access to many machine learning and artificial intelligence algorithms. Translating GMET to Python may make the software more accessible to many young researchers, so I believe that this work constitutes an interesting contribution to GMD.

I have, nonetheless, some comments and concerns about the current incarnation of the document that I would like to present.

Response: Thank you for the comprehensive review and insightful feedback on our manuscript. We have addressed each of your comments in detail below and revised the manuscript accordingly.

1. One of the main drawbacks of the new library, with respect to GMET, is its efficiency. It is a well-known drawback of Python, so it could have been expected. However, I wonder why, GMET being open source, the authors did not use the existing GMET FORTRAN code and wrap it using Cython, for instance, to get the best of both worlds. Many performant Python libraries, like SciPy, are just wrapping legacy FORTRAN code -for linear algebra, for instance-, making fast code available to Pythonistas. This approach would have saved time, since most of the routines were already programmed, and the effort would have been directed toward making the fast code accessible through Python. The total effort would have been smaller and the result equal on functional terms but better from a performance perspective.

Response: Thank you for your insightful feedback. We recognize the inherent performance advantages of Fortran, and the potential benefits of wrapping existing Fortran codes using tools like Cython. However, our primary goal with the development of the new package was not just to achieve a direct translation of GMET but to harness the extensive ecosystem, flexibility, and the rapid development pace that Python offers, especially with its growing capabilities in machine learning and data science.

While wrapping the Fortran codes would have indeed provided a performance boost, it would also come with its own set of challenges, such as increased complexity in maintenance, potential issues

with cross-platform compatibility, and limitations in seamlessly integrating with newer Python-based tools and libraries. By building directly in Python, we aimed to create a more accessible platform for the broader community, facilitating easier contributions, extensions, and integrations with other Python libraries. We do appreciate the value of performance, especially for large-scale applications, and will consider hybrid approaches cautiously in future versions, balancing both performance and development flexibility.

2. I miss some manual, or at the very least a small tutorial, to start to work with the library. Jupyter notebook is one of the standard ways to provide tutorials for Python libraries, which helps the user to familiarize with how to work with a new library in a graphical environment. I have checked the test cases in Zenodo, but they are quite a dry way to approach a new library. On top of showing its performance, I believe more developed examples would be very appreciated. Indeed, I believe the manual would be especially important to learn how to use the library in conjunction with sklearn since library interaction is not always an easy topic to grasp.

Response: The ./README.md and the ./docs/How_to_create_config_files.md serve as the manual for using the GPEP. We do recognize that the documents may not be so easy to follow for beginners. We have added a notebook, i.e., ./docs/GPEP_demo.ipynb, which streamlines downloading test cases from Zeono, run test cases, and visualize ensemble outputs. We have added clearer descriptions in Section 3.3. We thank the reviewer for taking the time to check the GMET cases.

3. A comment similar to the previous one, but related to the problem being solved. Existing users of GMET may already know the problem that the library solves, but new users could benefit from a small schematic definition of the problem. A small sketch or diagram could help the users to understand what problem the library solves and how it is solved. I have seen in the GitHub repo that there are many links to different papers, but a user, which may not necessarily be a scientist, may really appreciate a small theoretical introduction to the problem. In a sense, it would be making the library accessible for those outside academia that may use it to solve more practical problems.

With this comment in mind, I would say that section 2 could be extended so that readers may get a better idea of the library's inner workings. I believe that papers should provide enough detail to be self-contained, and I am not sure that all the most relevant details have been included in the document.

Response: Thank you for your suggestion. We agree that a more detailed introduction to the methodology will greatly benefit readers by providing all requisite information within the GPEP manuscript, minimizing the need to consult external literature. To this end, we have expanded Section 2 to include a more detailed exposition of the methodology. We have also segmented it into three distinct sub-sections for clearer understanding. While this provides a robust overview, we still advise readers seeking intricate technical details to refer to the original papers cited in Section 2, as we cannot introduce all nuances within this manuscript.

Besides, we also added a schematic in the manuscript, i.e., the new Figure 1, which shows how GPEP performs geospatial estimation.

Some minor comments:

Line 117, equation 1. I believe it should be two equations. It is just a problem with the organization of the equations.

Response: Thank you for pointing out this problem. Equation 1 should have two lines but the format was corrupted because of an unknown reason. We have fixed it.

Line 149: I think the 's' should be removed

Response: Yes. We have removed it.

---

## Author Comment (AC3)

This paper described a new Gridded Meteorological Ensemble Tool (GMET) for probabilistic hydro-meteorological estimations. The new tool was developed with Python, ensuring the improved flexibility and wider application ranges. This work would be a big contribution to the community for ensemble studies. However, there are a few concerns need to be addressed before the paper can be published.

Response: Thank you for the comprehensive review and insightful feedback on our manuscript. We have addressed each of your comments in detail below and revised the manuscript accordingly.

Input data intensity & model performance evaluation: The method relies heavily on available gauge data, and authors use US area as shown studies in this paper. I wonder how the method perform when applied to data-sparse area? LOO cross-validation method might not be enough, but author can evaluate for instance different proportion of calibration/validation gauges, to check the change in model performance.

Response: We agree that the data intensity has a notable impact on the accuracy of spatial estimation. However, the primary focus of this paper is on the software development, and the experiments in this study serve as demonstrations of the software's capabilities rather than a comprehensive assessment of the method's performance across different data scenarios. We show the cross validation results because it is the foundation of empirical/data-driven prediction and is provided as an intermediate output by GPEP.

The GPEP provides numerous spatial estimation choices, and their performance all depends on their data support, as with any numerical/statistical modeling effort. The GPEP itself, as a collection of various geospatial estimation methods, cannot be as easily quantitatively analyzed concerning its sensitivity to data intensity as would be a single-method software. While the topic of data intensity and its influence on model performance is important, it may be more suitable for a separate study that dives deeper into the methodological aspects.

L152, one of the largest advantages of translating GMET to GPEP is the easy extension of python libraries. For instance, the supervised learning method, however, authors didn't show how this could be operated in the current model or how results are differing compared to conventional methods.

Response: We have added an experiment using random forest in Section 4.2. Please see the new Figure 6 and relevant analyses. For the experiment, we only use the default settings of the sklearn package. The efficiency of random forest is influenced by factors like hyperparameters and feature combinations, but a deep dive into these is beyond the scope of this paper due to the rationale provided in our response to the previous comment.

Authors state that the spatial interpolation method could be different from investigated variables, however, details about the method selection and explanation are missing. I would like to know which method is more ideal for different basic variables and the reason.

Response: This problem you proposed is pertinent and scientifically challenging. In the context of this manuscript, we explore four variables: precipitation, Tmean, Trange, and SWE. GPEP provides many interpolation methods, ranging from global to local, and linear to non-linear regression techniques. There are also numerous parameters to account for, such as weight functions, the number of neighboring stations, transformation exponentials, and predictor combinations. These factors are intricately interlinked, adding to the complexity of the problem.

Furthermore, the ideal method may vary based on regional characteristics. This means that even with extensive research in this study in a specific region, the generalizability of our conclusions to other regions may be limited. Given these complexities, it's beyond the scope of this manuscript to recommend an optimal method for each basic variable.

However, we posit that GPEP is a promising tool to delve deeper into this scientific question. Given the depth and breadth of the subject, multiple papers would be required to comprehensively address and report the findings. The author team will no doubt investigate some aspects of this problem in the future, given that GPEP was developed to support new applications, but the purpose of this paper is to document the conversion of GMET and GPEP, describe and give examples of some of its features, rather than any detailed application-specific analysis of the ramifications of different available choices (such as interpolation method).

L315-316. Why using the standard deviation of KGE is better for temperature?

Response: This is because this method avoids the impact of units (e.g., Kelvin vs Celsius) and the amplified bias around zero temperature (when Celsius is used). We have added some explanation in the manuscript.

Figure 6. Why the uncertainty range is so large?

Response: A large uncertainty range arises when the statistical model does not capture strong relationships between the predictors and the predictands, thus the calibration and predictive errors of the relationship (e.g. regression) are relatively large. In this case, the predictands are daily weather parameters in complex, mountainous terrain, from the foothills of the eastern Rocky Mountains to the Continental Divide, which is a notoriously difficult estimation problem and a long-standing challenge in hydrometeorology. Estimating this extreme precipitation case (which occasioned the Boulder floods of 2013) using geospatial methods based on ground station networks, as depicted in Figure 7a (formerly Figure 6a), is particularly challenging. One of the strengths of probabilistic estimates is their ability to unveil this uncertainty through ensemble members. The large uncertainty range also means that a traditional deterministic estimation for the same event would likely be prone to substantial error (as is somewhat documented in the Gochis et al reference).

Figure 8. the scale of the two maps should be unified.

Response: We have unified their scale.

---

## Author Comment (AC4)

The manuscript touches upon an interesting and important subject, the creation of ensemble gridded data sets. The manuscripts presents a FORTRAN-to-Python translated, yet also improved, software package. In addition, the manuscript shows a number of example applications of the software. The manuscript could be a welcome addition to what is currently available to the scientific community.

The manuscript is written in a clear style, is interesting to read, and provides well-chosen examples.

Response: Thank you for the comprehensive review and insightful feedback on our manuscript. We have addressed each of your comments in detail below and revised the manuscript accordingly.

MAJOR COMMENTS

1) Given the importance of creating ensemble grids, which is reflected by the title and introduction, it would be desirable that the authors spend more time on explaining the way in which the software generates the ensemble members. Currently, the discussion of the ensemble creation methodology is not detailed enough.

Response: Thank you for emphasizing the significance of ensemble estimation. To address this, we have expanded Section 2 to provide more comprehensive explanations of both deterministic and probabilistic estimations, including both text descriptions and equations. We also divided Section 2 into three sub sections to be clearer. The updated methodology section now includes all essential introductions and has clearer organizations. Equations (5)-(7) provide a complete description of the probabilistic estimation.

Meanwhile, we still recommend readers consult the original documents cited within this section for in-depth understanding of intricate technical details such as the generation of SCRFs. The primary objective of this paper is to introduce the GPEP package, building upon established methodologies from GMET's legacy documents and codes. Echoing too many details that have been well documented in previous publications could distract from the contribution of this manuscript.

2) In the examples, there is much attention for the predictive accuracy of the ensemble mean. However, there is less emphasis on the accuracy of the ensemble dispersion. Given that one of the main pros for using this software is that it can create an ensemble data set, I would suggest that the authors show the accuracy of the gridded ensemble dispersion, for example by showing rank histograms.

Response: Thanks for the suggestion. The evaluation of probabilistic outputs will be a good demonstration for the GPEP. We have added the rank histogram and relevant analyses for precipitation, Tmean and Trange in Section 4.2.

3) The repository is somewhat minimal. Please consider including some example scripts, possibly in a Jupyter Notebook. Since the paper is essentially a presentation of new software, the repository could have some more features to introduce the software to new users.

Response: Yes, we have added a Jupyter Notebook that illustrates the steps to download, execute, and visualize the test cases. This notebook is designed to be user-friendly and is complemented by thorough explanations. Additionally, it provides pointers to more in-depth documentation when required. We also note that the conversion of GMET to GPEP represented a substantial amount of work that is documented by this paper, and the development of extended user support will naturally

be expected to grow as applications are undertaken. Over years of the existence and publications on GMET, user cases were only first introduced to the repository with GMET v2.0. GPEP has benefited from including some of those as a starting point.

MINOR COMMENTS

The text is clear and well-written. However, the figures require some more attention.

Fig. 1: Please make clear in the figure caption what LWR1, LWR2 and RF are. Also, in the main text, discuss why LWR2 is so much slower than LWR1.

Response: We have added explanations in the figure caption. The reason why LWR2 is slower than LWR1 is probably caused by factors such as the complexity and overhead of sklearn and the implementation difference (LWR1 is translated from Fortran codes using LU decomposition). We will investigate more in the future when optimizing the speed of GPEP. More discussions have been added in the main text.

Fig. 2: Please make sure to show axis lables and physical units in all figures.

Response: Done.

Fig. 3: Please make sure to show axis lables and physical units in all figures. The axis numbers on the x-axis are unreadable. Also, please make the color range of (c) and (d) the same.

Response: Done.

Fig. 4: Please make sure to show axis lables and physical units in all figures.

Response: Done.

Fig. 5: Please make sure to show axis lables and correct physical units in all figures. The axis numbers on the x-axis are unreadable.

Response: Done.

Fig. 6: It is unclear what is shown here; is this for a specific location, or is at a spatial average for a specific polygon? Please explain in figure caption.

Response: It is the spatial averaging for the latitude/longitude extents in the figure caption. We have added more explanation.

Fig. 7: Please make sure to show axis lables and physical units in all figures.

Response: Done.

Fig. 8: Please make color range of (b) and (c) correspond.

Response: Done.9

---

## Author Response (AR2)

The study presents a new open-source Python library for ensemble estimation of geospatial earth system variables. The new library (GPEP) is based on an existing one (GMET) that is programmed in Fortran. The authors aim to increase the flexibility that GMET provides, increasing the amount of variables that can be analyzed, the number of spatial interpolation schemes and other important characteristics. They apply the library to three demonstration experiments where they compare their results to those provided by GMET. They conclude by remarking the advantages of the new library and some of its drawbacks.

The main contribution of this work is to have translated a model in FORTRAN to Python. FORTRAN is an old-school programming language, fast and expressive, that has been extensively used for numerical model programming and other intensive tasks. It is less popular than it used to be and compiling it may be complicated. On the other hand, Python is an interpreted language that does not require to compile its codes, is very portable and with a plethora of libraries around it that automatically creates synergies with every new library, like it could be the case with GPEP.

The authors have addressed the comments that I had originally formulated in an acceptable way. I still would be missing a more user friendly tutorial, like, for instance, having used the Jupyter notebook to configure and comment the examples, instead of just downloading and plotting the results. In any case, the material they provide should be enough to start working with the library, although a bit more effort from the side of the use may be required. The only minor change that I would require, then would be to improve the documentation a little bit, with a Jupyter notebook that shows the creation of a case, how the information is recolected, etc. I believe they already have the information so it is just a matter of reorganizing things a bit.

In any case, I believe that this work constitutes an interesting contribution to GMD so my recommendation is to accept the paper.

Response: Thank you! We have added more descriptions of the test case in *./docs/ GPEP_demo.ipynb* and *./docs/How_to_create_config_files.md*. Those explain not only the configuration files, but also the files (e.g., netcdf and csv) that users need to create a case from scratch. Please see the latest pull request on Github: https://github.com/NCAR/GPEP/pull/8.

Besides, we would like to mention that the demonstration notebook *./docs/ GPEP_demo.ipynb* has already included test case running during our first round of revision. It runs the test cases for three experiments: (1) dynamic predictors using locally weighted regression, (2) static predictors using locally weighted regression, and (3) random forest global regression. The plotting in the notebook is based on the outputs that are generated by the users themselves on their laptops or servers instead of the downloaded data from Zenodo.

This revision adds more details to the Github repo, and no revision is made to the manuscript because Section "3.3 GPEP documentation and applicability" already provides a good description.

Anonymous Referee #3
Thanks for the authors' effort to improve the manuscript and the replies have sufficiently resolved my concerns. I'm satisfied with the updates and suggest it publish as its current form.
Response: Thank you!